# Acute effects of in-step and wrist weights on change of direction speed, accuracy and stroke velocity in junior tennis players

**Joshua Colomar** [1,2]*, **Ernest Baiget** [3], **Francisco Corbi**[4], **Joshua Muñoz**[1,2]

**1** National Institute of Sport and Physical Education (INEFC), University of Barcelona, Barcelona, Spain,
**2** Sports Science Department, Academia Sánchez-Casal, Barcelona, Spain, **3** Sport Performance Analysis Research Group (SPARG), University of Vic—Central University of Catalonia, Vic, Spain, **4** National Institute of Sport and Physical Education (INEFC), University of Lleida, Lleida, Spain

* joshuacolomar@gmail.com

## Abstract

The main aim of this study was to investigate the acute effects of the use of a weighting set (Powerinstep®) on measures of stroke velocity (StV), accuracy and change of direction speed (CODS) in junior tennis players. A within-subjects design was used to evaluate seventeen (6 female and 11 male) tennis players (mean ± SD; 16.5 ± 1.3 years old; 1.75 ± 8.4 m; 67.0 ± 8.1 kg; 22.04 ± 1.8 kg/m$^2$) on StV of three specific tennis actions (serve, forehand and backhand) and CODS for the following conditions: wearing a 50, 100, 150, 200 g weight or no weight at all (baseline). No significant differences were found between conditions for forehand (F = 0.412; p = 0.799), backhand (F = 0.269; p = 0.897) and serve (F = 0.541; p = 0.706) velocity and forehand (F = 1.688; p = 0.161), backhand (F = 0.567; p = 0.687) and serve (F = 2.382; p = 0.059) accuracy and CODS (F = 0.416; p = 0.797). Small-to-moderate effect sizes (ES) negatively affecting StV when using 200 g compared to the baseline (ES = 0.48, 0.35 and 0.45) could be observed. Moderate (ES = -0.49) and trivial (ES = -0.14 and -0.16) ES for a higher accuracy score were noticed in serve, forehand and backhand 100 g compared to the baseline. Moreover, small ES (ES = 0.41) for improvement in 200 g CODS comparing to baseline conditions were found. These results indicate that the use of a weighting set does not significantly affect StV or CODS respectively. Notwithstanding, small-to-moderate changes show impact in accuracy and no variance in velocity production when using 100 g alongside faster execution in CODS when using 200 g.

## Introduction

The development of hitting or throwing velocity in overhead sports has often involved improving movement patterns, enhancement of conditioning or modifying implement such as racquets or baseballs [1]. As speed, power and stroke velocity (StV) have become determinant factors of tennis [2,3], it may become interesting to observe specific strategies to improve velocity production that practitioners can use to manage and plan new training methods. Concerning modification of implement, and focusing on tennis, customizing racquets in order to alter their weight, balance point and swing weight is an extended practice performed by players

**Funding:** The research leading to these results has been conducted using funds from the agreement

between the Universitat de Vic – Universitat Central de Catalunya and Powerinstep, SL. The funders had no role in study design, data collection and analysis, decision to publish, or preparation of the manuscript.

**Competing interests:** The authors declare that there is no conflict of interest between the participants, the materials and equipment used, or any other procedure undertaken during the experiments and the researchers of this investigation.

and coaches [4]. This practice, in addition to other reasons, intends to use the transfer of momentum created by the mass of the racquet to hit the ball faster. In this line, heavier racquets will produce faster balls but consequently be swung slower than lighter versions [5]. Nevertheless, current literature is scarce about the effects of these variations and also offers doubts on how different customizing techniques (i.e., how the mass is distributed throughout the racquet) may affect speed [1,4] or accuracy. Moreover, intervention programs have suggested that the use of overweight implements or balls could be an effective way of improving throwing velocity in overhead sports [6,7] including tennis [8]. Although in this case tendencies have generally aimed to vary weight on the frame of the racquet, no investigations are available on how StV may be affected by the use of extra loading on extremities, raising uncertainty on how this may affect ball speed alongside kinetics and kinematics. Taking into account that the International Tennis Federation (ITF) does not prohibit the use of materials that modify the shape or physical properties of the racquet, the appearance of new equipment and training techniques may offer other ways of modifying momentum and consequently StV without modifying the racquet's features, giving insight on new ways of affecting velocity production. As a starting point, further knowledge on how StV and accuracy are affected when modifying swing weight could be interesting for developing specific intervention programs that seek to maximize the mechanical power output using light loads [9].

Added to this, around four changes of direction per point and as many as 1000 per match are produced during tennis match-play and cover on average a distance of 8–15 m per point [10,11], highlighting the importance of short distance sprinting, change of direction speed (CODS) and agility for competitive tennis players [3]. Following the aforementioned use of wearable resistance training systems in order to improve physical aspects in predominantly upper body actions, literature shows some interesting performance adaptations when using this kind of equipment. Aspects such as oxygen consumption or energy cost are increased when running using certain external light loads on the lower limbs [12]. Furthermore, the use of wearable devices on the trunk and limbs may also affect sporting aspects such as jumping and sprinting, decreasing or increasing performance [13,14]. The use of light loads that can easily be attached and don't interfere in the athlete's movement could enable higher execution velocities that performed in a sport-specific context may further optimize training adaptations [15]. However, literature seems to be limited when speaking of the effects of these wearables in over-the-ground sprinting or acyclic sporting actions such as agility or CODS [13], which would more appropriately fit those actions present in tennis match-play.

A mobile weighting set with the name of Powerinstep®, consisting of various weight capsules (50, 100, 150, 200 g) and a wristband or plastic pieces to place them on the player's wrist or instep could be one of the aforementioned systems that practitioners may be interested in using in order to develop velocity production on both, specific tennis strokes and change of direction performance. Therefore, the purpose of this study was to investigate the acute effects of the use of a weighting set (Powerinstep®) on the tennis player's wrist or shoe on measures of StV, accuracy and CODS in comparison with 5 different conditions (i.e., wearing 50, 100, 150, 200 g weights or no weights at all) in young competitive tennis players. It is hypothesized that the use of certain weights that increase the momentum of the swing without altering speed (i.e., 100 g and 150 g) and that do not exceed a certain weight and interfere in velocity production (i.e., 200 g) will improve StV without affecting accuracy. On the other hand, CODS will be negatively affected exponentially as weight increases.

## Materials and methods

### Subjects

Seventeen (6 female and 11 male) competitive tennis players (mean ± SD; age, 16.5 ± 1.3 years; height, 1.75 ± 8.4 m; weight, 67.0 ± 8.1 kg; BMI 22.04 ± 1.8 kg/m$^2$) with an International Tennis Number (ITN) ranging from 2 to 4 participated in this study. Based on the repeated-measures design and an anticipated statistical power of 0.80 with an effect size 1.2, it was determined that a minimal sample size of n = 15 subjects would be necessary (G-Power software version 3.1.9.5, University of Dusseldorf, Dusseldorf, Germany). The player's ITN was established by the consensus of three coaches accredited with RPT (Registry of Tennis Professionals) level 3, following the ITN Description of Standards. Out of the seventeen players, just one of them used a one-handed backhand style while the remaining subjects played two-handed. Participants had a weekly volume of training of 25h/week$^{-1}$, and were required to have a minimum of 1 year of experience in tennis and strength training. Also, they should not have experienced any pain in the trunk/upper body or other musculoskeletal discomfort in the six previous months.

### Ethics statement

All subjects were informed in advance about the characteristics of the study and, before their participation, the participants and their legal tutors, in the case of being underage, voluntarily signed an informed consent. The study was conducted following the ethical principles for biomedical research with human beings, established in the Declaration of Helsinki of the AMM (2013) and approved by the Ethics Committee of the Catalan Sports Council (01/2019/CEICEGC).

### Experimental design

A randomized, repeated measures within study design were assessed to compare the acute effects of wearing a set of weights (50, 100, 150, 200 g. Powerinstep®) with respect of not wearing them on StV, accuracy and CODS in young competitive tennis players. All weight sets were provided by Powerinstep® and consisted of one weight attached to a wristband for StV testing and two weights with instep plastic pieces for attachment to assess CODS (Fig 1). A familiarization session was carried out to inform on how to place the weights to avoid discomfort and possible inconveniences. Conditions were randomly distributed to avoid the influence of fatigue and test-learning effects. Subjects weren't familiarized with in-step or wrist weights. As dependent variables, StV (in km·h$^1$), accuracy points and CODS (in seconds) were recorded to compare between 4 different conditions (50, 100, 150, 200 g) and baseline conditions (0 g). The comparison between these situations aimed to investigate the effects of using light weight loads on StV, accuracy and CODS.

### Measurements

The collection of data took place in March during a normal in-season training week in groups of 4 players and on 2 separate testing sessions, performed in the morning and executed at least 48h apart. Participants hadn't trained in the previous 24h to any of the testing sessions and received all information regarding the risks and benefits of the study to obtain the informed consent in advance. Players were allowed to consume water ad libitum. Isotonic, energetic and caffeinated drinks were not allowed before or during the testing sessions. The first session consisted of performing the CODS test while the second session was scheduled to obtain StV and accuracy parameters.

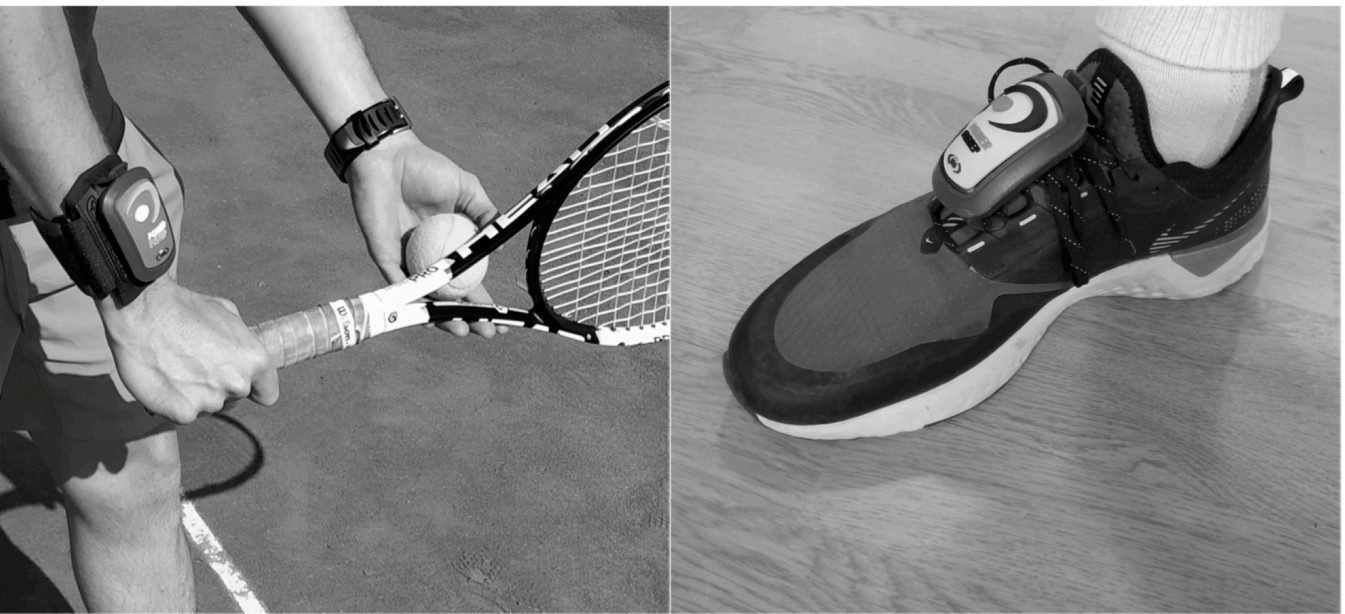

**Fig 1. Powerinstep® wristband and in-step weight attachment.**

**Maximum stroke velocity and accuracy.** Data collection was executed on a tennis hard court with stable wind conditions ($< 2 \, m \cdot s^1$) using new tennis balls (Head ATP Pro, Spain). Before the test, subjects performed a standardized warm-up that included mobility exercises, 5 minutes of free rallies and 5 to 10 progressive serves. Each subject randomly executed 5 series of 8 serves (4 on each side of the court) with 2 minutes of rest between sets for each one of the analyzed conditions (i.e., wearing a 50, 100, 150, 200 g or no weight set on the dominant wrist as shown in Fig 1). Following the serves, and after a 5-minute rest, participants performed 5 random series of 8 forehands and 8 backhands (crossed-court) without alternating strokes following each testing condition and following the same resting periods, as explained in Fig 2. Participants wore one of the weight sets exclusively attached to the dominant extremity. Only the serves that were in the serve box and the groundstrokes that landed in the singles court were registered. Maximum StV was determined using a hand-held radar gun (Stalker ATS II, USA, frequency: 34.7 GHz [Ka-Band] ± 50 MHz). The radar was positioned in the center of the baseline, 2 m behind the line and at an approximate height of 2 m for the serves and behind the player following the trajectory of the ball. Hitting as hard and precise as possible was indicated and immediate feedback was provided to the subjects to encourage maximum effort. To avoid variability performing groundstrokes, balls were fed by a ball-throwing machine (Pop-

| *Set 1* | *Rest* | *Set 2* | *Rest* | *Set 3* | *Rest* | *Set 4* | *Rest* | *Set 5* | |
|---|---|---|---|---|---|---|---|---|---|
| x8 Serves | 2min | x8 Serves | 2min | x8 Serves | 2min | x8 Serves | 2min | x8 Serves | |
| | | | | | | | | *Rest = 5min* | |
| x8 Forehands | 2min | x8 Forehands | 2min | x8 Forehands | 2min | x8 Forehands | 2min | x8 Forehands | |
| | | | | | | | | *Rest = 5min* | |
| x8 Backhands | 2min | x8 Backhands | 2min | x8 Backhands | 2min | x8 Backhands | 2min | x8 Backhands | |

**Fig 2. StV and accuracy experimental design.**

Lob Airmatic 104, France) at a constant speed (68.6 ± 1.9 km·h$^{-1}$). Also, accuracy of the strokes was registered for further analysis using a similar approach to Pialoux et al., 2015 [16] as explained in Fig 3. To assess serve accuracy, a ball that landed in the S1 area (1*1 m) accounted for 5 points; S2 (2*2 m), 3 points and S3 (remaining area of the serve box), 1 point. To assess groundstrokes, a ball that landed in the area FH1 or BH1 (2*2 m) accounted for 5 points; FH2 or BH2 (3*3 m), 3 points and FH3 or BH3 (rest of the tennis court besides doubles alleys), 1 point. All other ball placements resulted in zero points. Accuracy was defined by the sum of all points, with a higher score corresponding to a higher accuracy. StV assessment measurements showed good to excellent test-retest reliability (ICCs 0.73 to 0.96) with a coefficient of variation (CV) ranging from 4.6 to 5.9%. Accuracy showed poor to moderate test-restest reliability (ICCs <0.2 to 0.550), similar to previous investigations [17] but contrary to studies that found good reliability in similar assessments [18].

**CODS assessment.** To assess the ability to perform a single change of direction (CODS), the 505-agility test was performed on a tennis hard court [19]. Participants executed a standardized warm-up prior to the commencement of the test, consisting of a series of mobility exercises, a 5-minute jog and 3 progressive sprints. The 505-agility test consisted of sprinting from a standing position for 15 m (through the timing gates at 10 m) and executing a 180˚ change of direction on their preferred foot to further sprint through the timing gates [20]. Players assumed a preferred foot behind the starting position and started the test voluntarily. Results were registered using timing gates (Chronojump®, Barcelona, Spain), as they offer higher degrees of accuracy than stopwatch-recorded times [21]. All subjects executed the test two times with each one of the analyzed conditions (i.e., wearing a 50, 100, 150, 200 g on both feet (Fig 1) or no weight set in a randomized order. After every attempt, subjects were asked to rest for 1 minute prior to performing again. All measurements demonstrated a good to excellent test-retest reliability (ICCs 0.79 to 0.91) with CV ranging from 1.6 to 3.3%.

## Statistical analyses

Descriptive data were expressed as mean ± standard deviation (SD). The normality of the distributions and homogeneity of variances were assessed with the Shapiro–Wilk and Levene tests, respectively. The reliabilities of test measurements were assessed using intraclass correlation coefficients (ICCs), all of agility, serve, forehand and backhand velocity measurements reached an acceptable level of reliability (ICC > 0.73). The typical error of measurement (TEM) was calculated for the intraindividual test–retest strokes (i.e., forehand, backhand and service) and CODS variables and expressed as a mean CV. Differences between the StV and accuracy and CODS 0 g (baseline) and the scores at 4 conditions (50, 100, 150 and 200 g) were evaluated using a one-way analysis of variance (ANOVA) with repeated-measures with Bonferroni-corrected post hoc analysis. Mean differences in absolute and percent values were also used. The magnitude of the differences in mean was quantified as effect size (ES) and interpreted according to the criteria used by Cohen [22] (<0.2 = trivial, 0.2–0.5 = small, 0.5–0.8 = moderate, >0.8 = large). Because forehand velocity 0 g and 150 g data were not normally distributed, Friedman's test was used to examine the differences between baseline and different weights in forehand velocity. The level of significance was set at p ≤ 0.05. All statistical analyses were performed using SPSS 23.0 software (SPSS Inc., Chicago, IL, USA).

## Results

No significant differences were found between conditions for forehand (F = 0.412; p = 0.799), backhand (F = 0.269; p = 0.897) and serve (F = 0.541; p = 0.706) velocity and forehand

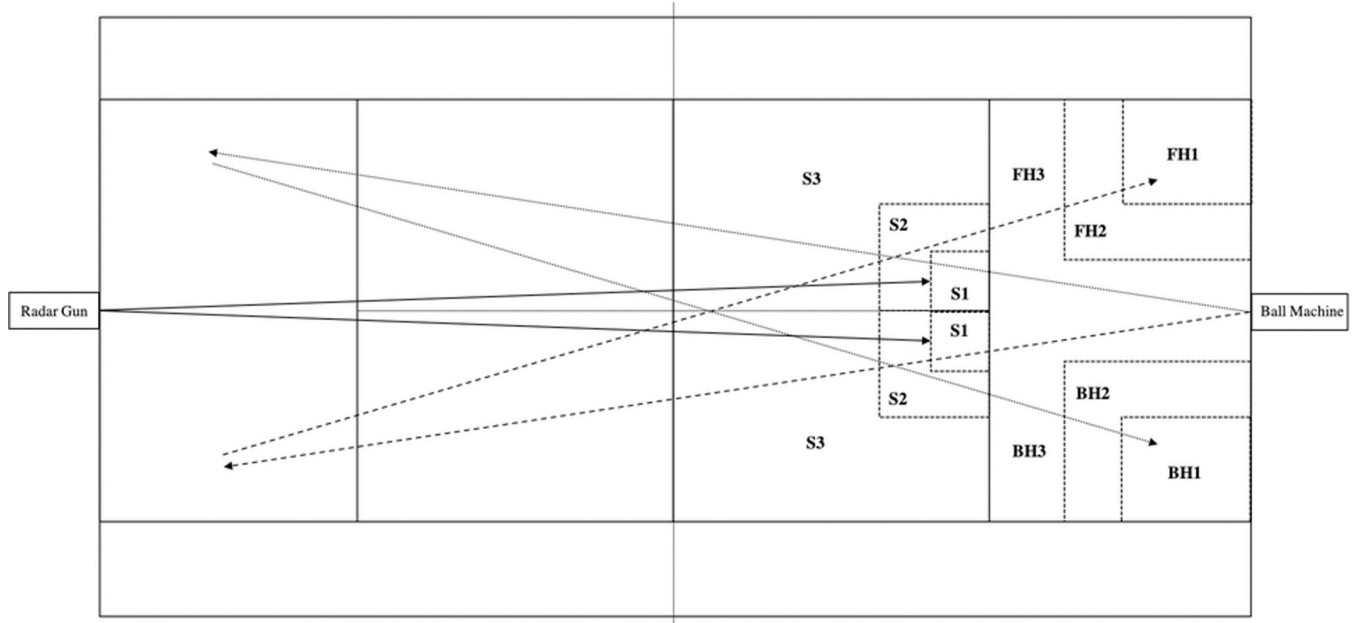

**Fig 3. Tennis court layout for stroke velocity and accuracy assessment.** S1, S2 and S3, the target areas for the serve; FH1, FH2 and FH3, the target areas for forehand drives; BH1, BH2 and BH3, the target areas for backhand drives. The full arrows indicate the ball trajectories for the serve, the dotted arrows indicate the ball trajectories for backhand drive, and the dash arrows the ball trajectories for forehand drive.

(F = 1.688; p = 0.161), backhand (F = 0.567; p = 0.687) and serve (F = 2.382; p = 0.059) accuracy and CODS (F = 0.416; p = 0.797).

There were no significant decreases and small-to-moderate effect sizes of StV in serve, forehand and backhand 200 g compared to the baseline (-4.5, -2.91 and -2.99%; ES = 0.48, 0.35 and 0.45) (Table 1). Moderate (23.04%; ES = -0.49) and trivial (6.06 and 7.33%; ES = -0.14 and -0.16) effect sizes for higher accuracy were found in serve, forehand and backhand 100 g compared to the baseline (Fig 4). A non-significant small effect size (-2.35%; ES = 0.41) for improvement in 200 g CODS comparing to the baseline conditions was observed (Fig 5).

**Table 1. Magnitude and percentage changes from baseline (0 g) in serve, forehand and backhand velocity and accuracy and change of direction speed (CODS) between 4 conditions (50, 100, 150 and 200 g).**

|  | 50 g | | 100 g | | 150 g | | 200g | |
|---|---|---|---|---|---|---|---|---|
|  | ES | % | ES | % | ES | % | ES | % |
| **Serve** | | | | | | | | |
| Velocity (km·h⁻¹) | 0.06 | -0.61 | 0.08 | -0.76 | 0.29 | -0.31 | 0.48 | -4.50 |
| Accuracy (points) | -0.08 | 4.30 | -0.49 | 23.04 | 0.55 | -29.58 | 0.11 | -5.42 |
| **Forehand** | | | | | | | | |
| Velocity (km·h⁻¹) | -0.06 | 0.52 | 0.06 | -0.50 | -0.01 | 0.10 | 0.35 | -2.91 |
| Accuracy (points) | 0.48 | -23.40 | -0.14 | 6.06 | 0.25 | -10.94 | 0.53 | -21.47 |
| **Backhand** | | | | | | | | |
| Velocity (km·h⁻¹) | -0.05 | 0.36 | 0.13 | -0.96 | 0.02 | -0.14 | 0.45 | -2.99 |
| Accuracy (points) | 0.00 | 0.00 | -0.16 | 7.33 | -0.01 | 0.44 | 0.31 | -12.09 |
| **CODS** | | | | | | | | |
| Time (s) | 0.13 | -0.60 | 0.13 | -0.64 | 0.11 | -0.48 | 0.41 | -2.35 |

ES, Cohen's effect size; CODS, change of direction speed.

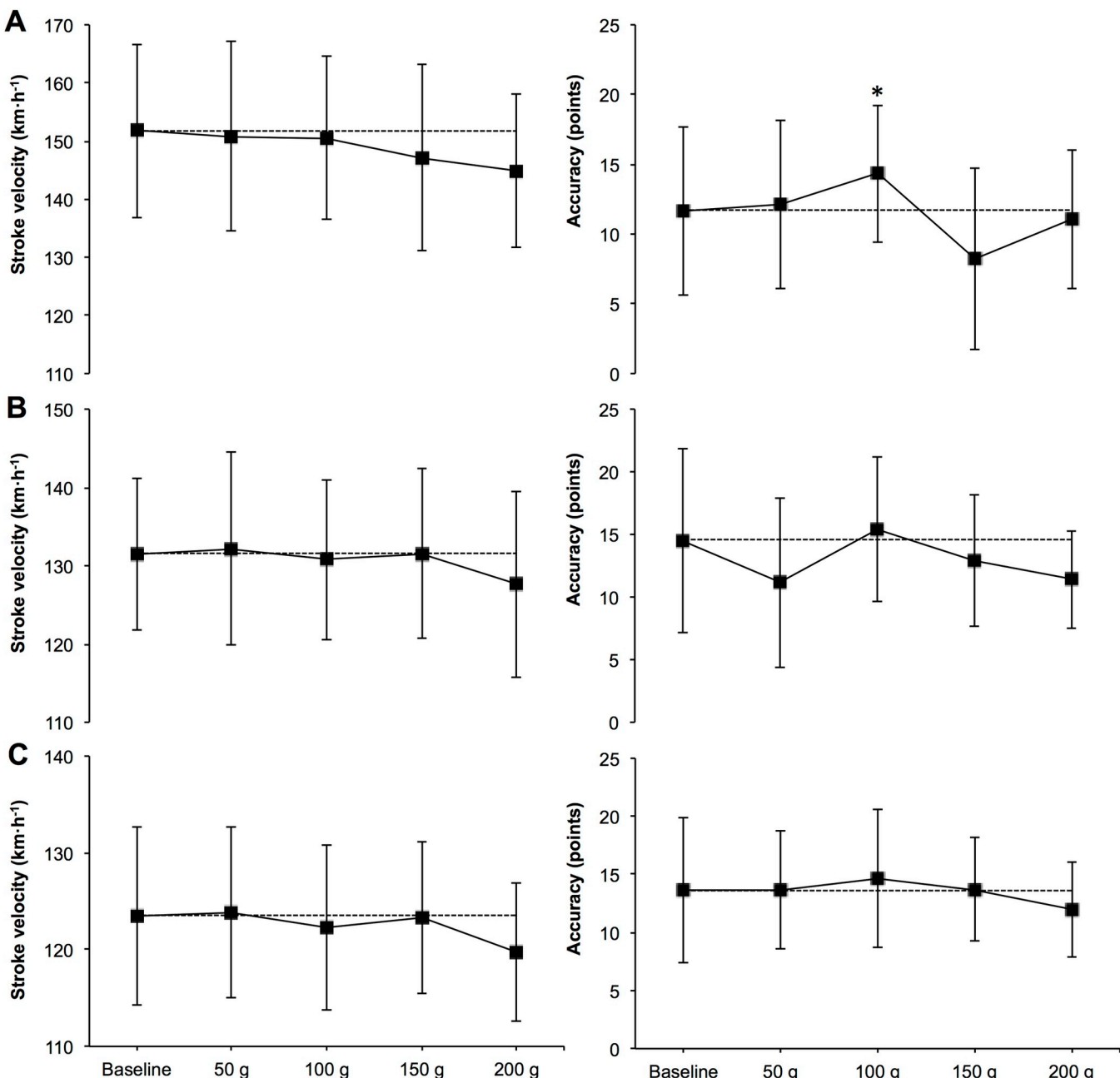

**Fig 4.** Comparisons of serve (A), forehand (B) and backhand (C) velocity and accuracy between 4 conditions (50, 100, 150 and 200 g). *Significant change from 150 g at p ≤ 0.05.

## Discussion

The main findings of this investigation were that the use of external light loads on upper and lower extremities do not seem to have significant effects on StV or CODS in junior tennis players. However, certain negative small-to-moderate changes were observed regarding StV when using heavier loads (200 g) and a higher accuracy without affecting velocity when using moderate loads (100 g). Regarding the use of weights on lower limbs, similar changes indicated that the use of heavier loads (200 g) could affect CODS in a positive way. Although no significant

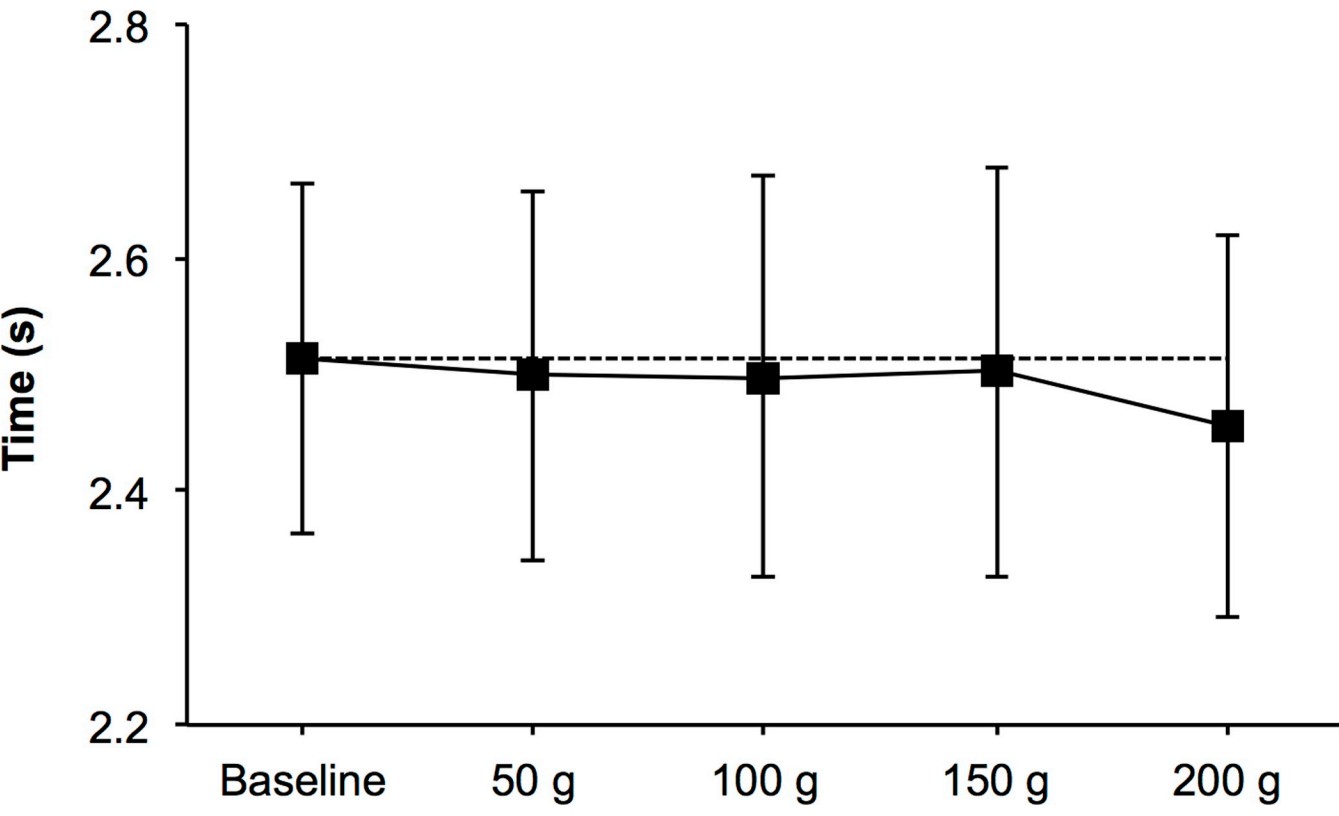

**Fig 5. Comparisons of change of direction speed (CODS) between 5 conditions (0, 50, 100, 150 and 200 g).**

increases in performance were observed by using a weighting set, no variables were diminished either.

More specifically, the lack of significant positive results regarding higher StV when using certain weights matches findings in other similar studies [1,4]. While literature has mainly focused on the acute effects of serve speed when adding weight to the racquet rather than the extremity as in this study, results did not find significant increases in velocity either. Even though a higher momentum caused by a heavier extremity could result in greater StV, the need of maintaining an optimal speed of the swing is necessary to benefit from this principle. As suggested by other authors, an increment in weight might cause deceleration in key determinant contributors to velocity production as internal rotation speed of the arm in the case of the serve [1,4,23]. Moreover, heavier loads placed on the extremity instead of the implement could reduce the speed of the racquet head due to a decreased linear and angular speed of the wrist, which is an important contributor to velocity production [23]. Precisely this issue may be the causative of no increases in speed in any of the weights used in this investigation and the greater loss of velocity that seems to happen when using 200 g weights (Fig 4). Interestingly, and focusing on groundstrokes, similar changes towards a decrease in StV occurred in players with a 2-handed backhand and the single subject that performed a 1-handed backhand with the weight and wrist band on his dominant extremity (2.99 and 1.12%; ES = 0.45 and ES = 0.49 respectively). Differences in both types of strokes rely on aspects such as a greater trunk rotation in the 2-handed backhand and a more rotated shoulder complex when playing with one hand [24]. In any case, as literature points out, players with either technique are able to produce similar horizontal racquet speed relying on a higher linear velocity in the 1-handed

fashion or angular velocity in the 2-handed style [24]. The fact that two strokes that build speed around different kinematic aspects but obtain similar results when performing with extra light loading as in this study, may reinforce the idea that certain weights affect key factors that influence the players ability to provide speed to the stroke. Added to this, investigations have found important kinematic and physical differences between elite and competitive players, concluding that those of a greater level rely on certain variables to produce speed. Aspects such as a more efficient use of elastic energy in leg extensors [2] or horizontal shoulder and racquet velocities [25], among others, contribute to enhancing StV, highlighting the importance of specific strength and kinematic parameters. As stated previously, the use of weights on the player's extremity may affect some of the mentioned key factors. Moreover, only players of a certain age and level may be able to maintain arm and racquet swing speed invariable and benefit from a higher momentum at impact on both, groundstrokes and serves. As a limitation of this study and aspects further investigations could focus on, the analysis of kinematic differences between the use of different weights and maturity/age status differences of the players could be registered to offer a further approach to the results obtained. Regarding the differences observed in the use of moderate weights (100 g), results seem to indicate slight changes towards an increased accuracy with unaffected velocity. It may appear that this could be a suitable load to observe positive longitudinal effects on StV or accuracy. Unlike non-significant immediate results observed in investigations that focused on acute effects [1,4], longitudinal studies that proposed the use of extra light loading around the implement or mobile offered positive increases in other overhead sports [6,7] besides tennis [8]. As literature suggests, the use of these kinds of strength training programs seem to be a good way of enhancing velocity production [26], benefiting from the principle of overload. On the other hand, this approach could compromise other factors such as kinematics and kinetics of the sporting action or injury rates [7]. Following suggestions presented by other authors, these interventions could be a way of improving velocity production after achieving a certain strength level in previous programs to, after, transfer these gains into specific tennis actions such as the serve and groundstrokes [4]. Concerning accuracy, as results seem to show small-to-moderate differences for greater scores with velocity unaffected, the use of this approach to training may offer players and coaches some beneficial technical outcomes regarding skill acquisition based on variability during the training of the stroke itself, following modern coaching practices [27]. In any case, to our knowledge, this is the first study to examine the acute effects of increasing weight on extremities) on StV or accuracy, manifesting the need of further investigations to expose such statements. As a limitation, and regarding accuracy reliability, the test was probably limited by asking subjects to hit the ball at maximum speed, causing greater variability in accuracy and consequently decreasing it. This issue has previously been observed in tennis [17] and is frequent when testing accuracy.

Regarding the use of light weights on lower limbs, no studies, to our knowledge, have attempted to investigate the effects on agility aspects or, more specifically, on CODS. Linear sprinting has received attention from literature both on treadmill and over-the-ground conditions [13,28], showing no changes in running or sprinting technique but decreases in performance (maximum sprint running), especially in the acceleration phase due to a reduction in stride frequency [29]. Contrary to results noticed when analyzing StV, the differences observed in this study showed a small decrease in time when using the heavier load (200 g), unlike the mentioned researches. These contrary results could be due to the differences in the weight used in previous investigations. The loads presented ranged from 1–5% of bodyweight in the mentioned studies whereas the higher load in this investigation (i.e., 200 g) accounted for around 0.335% of bodyweight. Loads of a certain magnitude may interpose stride frequency and consequently sprinting velocity. Although little literature is available on this matter,

presumably we will find differences when analyzing linear sprinting and change of direction or agility parameters such as the present here. In fact, some authors have analyzed kinematic factors affecting CODS and found better performances in those groups that had an increased stride frequency [30]. The use of wearable weights may cause greater stride rate triggered by the enhanced gravitational forces [31] and consequently result beneficial for agility-based tasks as the 505-agility test analyzed in this study. At any rate, further studies should focus on investigating longitudinally the effects of in-step weights on change of direction and agility and examine how loads may affect essential kinematic aspects such as stride length or frequency that are key determinants of CODS [13] performance before being able to state this.

In conclusion, the use of a weighting set on both wrists and in-steps does not significantly affect StV or CODS respectively. Although differences are not observed, the use of these light weights do not affect negatively velocity production or accuracy scores in junior tennis players either. Taking into account that further investigation is needed, small-to-moderate differences show an interesting improvement in accuracy and no variance in velocity production when using some of the weights tested (i.e., 100 g), suggesting that the use of this kind of apparel as a training tool could result in some way useful. This study also shows certain small changes for an increased performance in CODS when using 200 g in-step weights, suggesting that gear of these characteristics may affect change of direction or agility aspects to some extent. In any case, further investigations on the effects of the use of weighting sets on StV and CODS would be of great interest.

## Practical applications

Taking into account that using certain external light loads on the upper limbs in the form of a weight set does not seem to affect negatively velocity production or accuracy scores in young competitive tennis players, the use of this kind of apparel as a training tool could result in improvements on StV in the mid-long term, as suggested in similar literature [8]. Most likely, it would be preferable that strength training preceded wearable weight interventions, being this type of protocols more adequate for in-season programs where the goal is to transfer strength gains into specific tennis actions. Furthermore, programs should be applied with caution and not be maintained during long periods of training or competition since some studies suggest compromised kinematics and kinetics of the sporting action or increases in injury rates when analyzing light-weight interventions [7]. Moreover, variability of practice may be induced by the use of this piece of equipment and offer coaches and players new insights in emergent methods of training [27]. Regarding the use of in-step weights and their effects on CODS, further studies are needed to examine how loads may affect essential kinematic aspects such as stride length or frequency that are key determinants of CODS performance.

## Acknowledgments

The authors thank all the players and coaches for their enthusiastic participation. They would like to thank Academia Sánchez-Casal Barcelona. The research leading to these results has been conducted using funds from the agreement between the Universitat de Vic–Universitat Central de Catalunya and Powerinstep, SL. The authors declare that there is no conflict of interest between the participants, the materials and equipment used, or any other procedure undertaken during the experiments and the researchers of this investigation.

## Author Contributions

**Conceptualization:** Joshua Colomar, Ernest Baiget, Francisco Corbi.

**Data curation:** Ernest Baiget.

**Formal analysis:** Ernest Baiget.

**Funding acquisition:** Ernest Baiget.

**Investigation:** Joshua Colomar, Joshua Muñoz.

**Resources:** Joshua Muñoz.

**Supervision:** Francisco Corbi.

**Validation:** Joshua Colomar, Ernest Baiget, Francisco Corbi.

**Visualization:** Joshua Colomar.

**Writing – original draft:** Joshua Colomar.

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
