## [Decision Letter · Decision Letter 0]

5 Feb 2020

PONE-D-19-30941

Acute effects of in-step and wrist weights on change of direction and stroke velocity in junior tennis players

PLOS ONE

Dear Mr. Colomar,

Thank you for submitting your manuscript to PLOS ONE. After careful consideration, we feel that it has merit but does not fully meet PLOS ONE’s publication criteria as it currently stands. Therefore, we invite you to submit a revised version of the manuscript that addresses the points raised during the review process.

Please address the reviewers comments in a point by point manner.

We would appreciate receiving your revised manuscript by Mar 21 2020 11:59PM. To enhance the reproducibility of your results, we recommend that if applicable you deposit your laboratory protocols in protocols.io, where a protocol can be assigned its own identifier (DOI) such that it can be cited independently in the future. For instructions see: http://journals.plos.org/plosone/s/submission-guidelines#loc-laboratory-protocols

We look forward to receiving your revised manuscript.

Kind regards,

Caroline Sunderland

Academic Editor

PLOS ONE

Journal Requirements:

Reviewers' comments:

Reviewer's Responses to Questions

**Comments to the Author**

1. Is the manuscript technically sound, and do the data support the conclusions?

Reviewer #1: Yes

Reviewer #2: Yes

2. Has the statistical analysis been performed appropriately and rigorously? 

Reviewer #1: Yes

Reviewer #2: No

3. Have the authors made all data underlying the findings in their manuscript fully available?

Reviewer #1: Yes

Reviewer #2: Yes

4. Is the manuscript presented in an intelligible fashion and written in standard English?

Reviewer #1: Yes

Reviewer #2: Yes

5. Review Comments to the Author

Reviewer #1: Dear Authors,

The research is original and is designed to solve a problem that requires a solution in the field.

Title: The authors identified the title of the research in accordance with the content of the study.

Introduction: They have explained the scientific basis of the research well, but there is a need for a kinematic explanation of accelerating the racket head in tennis.

Method: The method determined by the authors in accordance with the hypothesis of the study is appropriate. As a Tennis Performance Coach, tennis coach and academician working in this field, it is important to remember that free weights can slow down the linear and angular speed of the wrist instead of speeding up the racket head. It can be thought that the athlete has made this effect unconsciously in order to minimize the risk of injury in eccentric control during the follow throuh phase. In addition, gravital resistance may negatively affect the horizontal force production at that time. Short-term tests may not be a problem, but the player's technique may change for long-term training periods. In particular, a negative change in the technique of an athlete trying to climb to the top of the performance is something we do not want to see as a coach.

Results: The data obtained in the study were presented with an appropriate statistical analysis.

Discussion: It is useful to explain why there is no significant relationship between the findings of this section. In this sense, it is useful to express my concerns in the method section.

Reviewer #2: General comments:

The main aim of this study was to investigate the acute effects of the use of a weighting set (Powerinstep®) on measures of stroke velocity (StV), accuracy and change of direction speed (CODS) in junior tennis players. It was a within-subjects design with seventeen young tennis players. It was adopted five experimental conditions/treatments (50, 100, 150, 200 g or no weight) for evaluate the effect on stroke velocity of three specific tennis actions (serve, forehand and backhand), accuracy, and change of direction speed. It's about a good and original study that show new results for scientific literature. However, the paper need some adjustments so that it can be published in Plos One.

Specific comments:

Title - It is recommended “Acute effects of in-step and wrist weights on change of direction speed, accuracy and stroke velocity in junior tennis players"

Abstract

Lines 26-29. The Anova results (F and p) could be descriptive here before ES. For exemple, “No significant differences (p > 0.05) were found between conditions for accuracy”

Materials and methods

Was the biologic maturation measured (e.g., maturity-off-set or sexual maturity)? In case positive, it is suggested that biological maturation is statistically controlled in the data analysis. In case negative, it is recommended indicating (discussion) as a limitation.

Subjects

Lines 92-94. Was it performed sample calculated for study? Considering the five experimental conditions/treatments, perhaps the 17 young participants was no enough for statistical analysis. It is recommended conduct a-priori sample size or a-posteriori sample size (i.e., power analysis).

Line 96: “Registro Profissional de Tênis” could be written in English.

Experimental design

Lines 118-123. How much washout (e.g., 24-h, 1-week) was adopted for different experimental conditions?

Lines 120-121. What was the sequence these tests? Was it randomized? How much time of rest/interval between tests?

Lines 112-114. Do players ingested caffeine or some ergogenic substance before experimental conditions visits? This information is important.

“Maximum stroke velocity and accuracy”

Lines 164-165 - It is suggested to quote study that has found good reliability to the serve, forehand or backhand accuracy.

https://www.rpd-online.com/article/view/v28-n1-desousa-sousa-andrade-etal

Guillot, A., Di Rienzo, F., Pialoux, V., Simon, G., Skinner, S., and Rogowski, I. (2015). Implementation of motor imagery during specific aerobic training session in young tennis players. Plos One, 10(11), e0143331. doi:10.1371/journal.pone.0143331

Hayes, M. J., Spits, D. R., Watts, D. G., and Kelly, V. G. (2018). The relationship between tennis serve velocity and select performance measures. Journal of Strength and Conditioning Research, a head of print. doi: 10.1519/JSC.0000000000002440

Statistical analysis

Line 204 - Remove “1988”

Results

Lines 210-211. Could be indicated all p values for comparisons (stV, accuracy and CODS), as well as F and ES for each comparison.

6. PLOS authors have the option to publish the peer review history of their article (what does this mean?). If published, this will include your full peer review and any attached files.

Reviewer #1: Yes: Suat YILDIZ, PhD.

Reviewer #2: Yes: Leonardo de Sousa Fortes

---

## [Author Response · Author response to Decision Letter 0]

13 Feb 2020

Reviewer #1:

Dear Authors,

The research is original and is designed to solve a problem that requires a solution in the field.

First of all, the authors would like to thank the reviewer for the comments and insights on this manuscript.

Title: The authors identified the title of the research in accordance with the content of the study.

Introduction: They have explained the scientific basis of the research well, but there is a need for a kinematic explanation of accelerating the racket head in tennis.

Method: The method determined by the authors in accordance with the hypothesis of the study is appropriate. As a Tennis Performance Coach, tennis coach and academician working in this field, it is important to remember that free weights can slow down the linear and angular speed of the wrist instead of speeding up the racket head. It can be thought that the athlete has made this effect unconsciously in order to minimize the risk of injury in eccentric control during the follow throuh phase. In addition, gravital resistance may negatively affect the horizontal force production at that time. Short-term tests may not be a problem, but the player's technique may change for long-term training periods. In particular, a negative change in the technique of an athlete trying to climb to the top of the performance is something we do not want to see as a coach.

Comment acknowledged. To respond to the reviewers concerns, we added the following statement in the introduction section to restate the little knowledge on how this type of equipment might affect technique and injury rate in tennis players: ‘Although in this case tendencies have generally aimed to vary weight on the frame of the racquet, no investigations are available on how StV may be affected by the use of extra loading on extremities, raising uncertainty on how this may affect ball speed alongside kinetics and kinematics’. Page 3, lines 52-55.

Moreover, and to emphasize the fact that this equipment is most likely useful for certain moments of the tennis session or fitness programs and not yet studied in the long-term, we added the following information to the practical applications section: ‘Furthermore, programs should be applied with caution and not be maintained during long periods of training or competition since some studies suggest compromised kinematics and kinetics of the sporting action or increases in injury rates when analyzing light-weight interventions [7]’. Page 16, lines 358-361.

Results: The data obtained in the study were presented with an appropriate statistical analysis.

Discussion: It is useful to explain why there is no significant relationship between the findings of this section. In this sense, it is useful to express my concerns in the method section.

Comment acknowledged. To support the reasons presented to explain the non-significant findings, we added information on what the reviewer pointed out. The heavier weights placed on extremities could decrease angular and linear speed and therefore not be able to use a greater momentum provided by the extra loading. We added the following sentence to the discussion section: ‘Moreover, heavier loads placed on the extremity instead of the implement could reduce the speed of the racquet head due to a decreased linear and angular speed of the wrist, which is an important contributor to velocity production [23]’. Page 12, lines 263-266.

Reviewer #2:

General comments:

The main aim of this study was to investigate the acute effects of the use of a weighting set (Powerinstep®) on measures of stroke velocity (StV), accuracy and change of direction speed (CODS) in junior tennis players. It was a within-subjects design with seventeen young tennis players. It was adopted five experimental conditions/treatments (50, 100, 150, 200 g or no weight) for evaluate the effect on stroke velocity of three specific tennis actions (serve, forehand and backhand), accuracy, and change of direction speed. It's about a good and original study that show new results for scientific literature. However, the paper need some adjustments so that it can be published in Plos One.

First of all, authors would like to thank the reviewer for the comments and insights on this manuscript.

Specific comments:

Title - It is recommended “Acute effects of in-step and wrist weights on change of direction speed, accuracy and stroke velocity in junior tennis players"

Comment acknowledged. The title was changed to; ‘Acute effects of in-step and wrist weights on change of direction speed, accuracy and stroke velocity in junior tennis players’. Page 1, lines 1-3.

Abstract

Lines 26-29. The Anova results (F and p) could be descriptive here before ES. For exemple, “No significant differences (p > 0.05) were found between conditions for accuracy”

Comment acknowledged. ANOVA results were added and are descriptive for all variables measured. The lines were rephrased to; ‘No significant differences were found between conditions for forehand (F = 0.412; p = 0.799), backhand (F = 0.269; p = 0.897) and serve (F = 0.541; p = 0.706) velocity and forehand (F = 1.688; p = 0.161), backhand (F = 0.567; p = 0.687) and serve (F = 2.382; p = 0.059) accuracy and CODS (F = 0.416; p = 0.797)’. Page 2, lines 23-27.

Materials and methods

Was the biologic maturation measured (e.g., maturity-off-set or sexual maturity)? In case positive, it is suggested that biological maturation is statistically controlled in the data analysis. In case negative, it is recommended indicating (discussion) as a limitation.

Comment acknowledged. Maturity status was not measured. As suggested, this was added as a limitation of the study and aspects further investigations could focus on. ‘As a limitation of this study and aspects further investigations could focus on, the analysis of kinematic differences between the use of different weights and maturity/age status differences of the players could be registered to offer a further approach to the results obtained’. Page 13, lines 288-291.

Subjects

Lines 92-94. Was it performed sample calculated for study? Considering the five experimental conditions/treatments, perhaps the 17 young participants was no enough for statistical analysis. It is recommended conduct a-priori sample size or a-posteriori sample size (i.e., power analysis).

Comment acknowledged. The power analysis was added to the participants section; ‘Based on the repeated-measures design and an anticipated statistical power of 0.80 with an effect size 1.2, it was determined that a minimal sample size of n = 15 subjects would be necessary (G-Power software version 3.1.9.5, University of Dusseldorf, Dusseldorf, Germany)’. Page 5, lines 97-101.

Line 96: “Registro Profissional de Tênis” could be written in English.

Comment acknowledged. Rephrased to ‘Registry of Tennis Professionals’. Page 5, line 102.

Experimental design

Lines 118-123. How much washout (e.g., 24-h, 1-week) was adopted for different experimental conditions?

Comment acknowledged. This is indicated in page 7, lines 137-138: ‘Participants hadn’t trained in the previous 24h to any of the testing sessions’. Nevertheless, the following information was also added to the measurements section: ‘The collection of data took place in March during a normal in-season training week in groups of 4 players and on 2 separate testing sessions, performed in the morning and executed at least 48h apart’. Page 7, lines 135-137.

Lines 120-121. What was the sequence these tests? Was it randomized? How much time of rest/interval between tests?

The information on the sequence of testing is provided in the ‘maximum stroke velocity’ and ‘CODS assessment’ sections since in our opinion it is more clear to the reader. Also, this information is supported by figure 2.

‘Each subject randomly executed 5 series of 8 serves (4 on each side of the court) with 2 minutes of rest between sets for each one of the analyzed conditions (i.e., wearing a 50, 100, 150, 200 g or no weight set on the dominant wrist as shown in Figure 1). Following the serves, and after a 5-minute rest, participants performed 5 random series of 8 forehands and 8 backhands (crossed-court) without alternating strokes following each testing condition and following the same resting periods, as explained in Figure 2’. Page 7, lines 148-154.

‘All subjects executed the test two times with each one of the analyzed conditions (i.e., wearing a 50, 100, 150, 200 g on both feet (Figure 1) or no weight set in a randomized order. After every attempt, subjects were asked to rest for 1 minute prior to performing again’. Page 9, lines 194-197.

If the reviewer considers this information relevant to be included in the ‘experimental design’ or ‘measurements’ section, it could be changed. In any case, we consider it easier to follow for the reader with this distribution.

Lines 112-114. Do players ingested caffeine or some ergogenic substance before experimental conditions visits? This information is important.

Comment acknowledged. Information on this regard was added. ‘Players were allowed to consume water ad libitum. Isotonic, energetic and caffeinated drinks were not allowed before or during the testing sessions’. Page 7, lines 139-141.

“Maximum stroke velocity and accuracy”

Lines 164-165 - It is suggested to quote study that has found good reliability to the serve, forehand or backhand accuracy.

https://www.rpd-online.com/article/view/v28-n1-desousa-sousa-andrade-etal

Guillot, A., Di Rienzo, F., Pialoux, V., Simon, G., Skinner, S., and Rogowski, I. (2015). Implementation of motor imagery during specific aerobic training session in young tennis players. Plos One, 10(11), e0143331. doi:10.1371/journal.pone.0143331

Hayes, M. J., Spits, D. R., Watts, D. G., and Kelly, V. G. (2018). The relationship between tennis serve velocity and select performance measures. Journal of Strength and Conditioning Research, a head of print. doi: 10.1519/JSC.0000000000002440

Comment acknowledged. A reference was added and we rephrased to: ‘Accuracy showed poor to moderate test-restest reliability (ICCs <0.2 to 0.550), similar to previous investigations [17] but contrary to studies that found good reliability in similar assessments [18]’. Page 8, lines 173-175.

Statistical analysis

Line 204 - Remove “1988”

Comment acknowledged. Year was removed.

Results

Lines 210-211. Could be indicated all p values for comparisons (stV, accuracy and CODS), as well as F and ES for each comparison.

Comment acknowledged. The following paragraph was added to the results section; ’No significant differences were found between conditions for forehand (F = 0.412; p = 0.799), backhand (F = 0.269; p = 0.897) and serve (F = 0.541; p = 0.706) velocity and forehand (F = 1.688; p = 0.161), backhand (F = 0.567; p = 0.687) and serve (F = 2.382; p = 0.059) accuracy and CODS (F = 0.416; p = 0.797)’. Page 10, lines 220-223. 

ES for each comparison is indicated in Table 1.

---

## [Decision Letter · Decision Letter 1]

5 Mar 2020

Acute effects of in-step and wrist weights on change of direction speed, accuracy and stroke velocity in junior tennis players

PONE-D-19-30941R1

Dear Dr. Colomar,

We are pleased to inform you that your manuscript has been judged scientifically suitable for publication and will be formally accepted for publication once it complies with all outstanding technical requirements.

With kind regards,

Caroline Sunderland

Academic Editor

PLOS ONE

Additional Editor Comments (optional):

Reviewers' comments:

Reviewer's Responses to Questions

**Comments to the Author**

1. If the authors have adequately addressed your comments raised in a previous round of review and you feel that this manuscript is now acceptable for publication, you may indicate that here to bypass the “Comments to the Author” section, enter your conflict of interest statement in the “Confidential to Editor” section, and submit your "Accept" recommendation.

Reviewer #1: All comments have been addressed

Reviewer #2: All comments have been addressed

2. Is the manuscript technically sound, and do the data support the conclusions?

Reviewer #1: Yes

Reviewer #2: Yes

3. Has the statistical analysis been performed appropriately and rigorously? 

Reviewer #1: Yes

Reviewer #2: Yes

4. Have the authors made all data underlying the findings in their manuscript fully available?

Reviewer #1: Yes

Reviewer #2: Yes

5. Is the manuscript presented in an intelligible fashion and written in standard English?

Reviewer #1: Yes

Reviewer #2: Yes

6. Review Comments to the Author

Reviewer #1: (No Response)

Reviewer #2: Congratulations to the authors for the changes. The paper's improved considerably. I consider the paper to be accepted to publication in "Plos One".

7. PLOS authors have the option to publish the peer review history of their article (what does this mean?). If published, this will include your full peer review and any attached files.

Reviewer #1: No

Reviewer #2: Yes: Leonardo de Sousa Fortes

---

## [Editor Report · Acceptance letter]

9 Mar 2020

PONE-D-19-30941R1 

Acute effects of in-step and wrist weights on change of direction speed, accuracy and stroke velocity in junior tennis players 

Dear Dr. Colomar:

I am pleased to inform you that your manuscript has been deemed suitable for publication in PLOS ONE. Congratulations! Your manuscript is now with our production department. 

With kind regards,

on behalf of

Dr. Caroline Sunderland 

Academic Editor

PLOS ONE